# Heavy Metals Removal from Water by Efficient Adsorbents

**Muhammad Zaim Anaqi Zaimee, Mohd Sani Sarjadi** 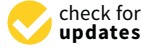 **and Md Lutfor Rahman ***

Industrial Chemistry Program, Faculty of Science and Natural Resources, Universiti Sabah Malaysia, Kota Kinabalu 88400, Malaysia; zaimzaimee98@gmail.com (M.Z.A.Z.); msani@ums.edu.my (M.S.S.)
* Correspondence: lotfor@ums.edu.my

**Abstract:** Natural occurrence and anthropogenic practices contribute to the release of pollutants, specifically heavy metals, in water over the years. Therefore, this leads to a demand of proper water treatment to minimize the harmful effects of the toxic heavy metals in water, so that a supply of clean water can be distributed into the environment or household. This review highlights several water treatment methods that can be used in removing heavy metal from water. Among various treatment methods, the adsorption process is considered as one of the highly effective treatments of heavy metals and the functionalization of adsorbents can fully enhance the adsorption process. Therefore, four classes of adsorbent sources are highlighted: polymeric, natural mineral, industrial by-product, and carbon nanomaterial adsorbent. The major purpose of this review is to gather up-to-date information on research and development on various adsorbents in the treatment of heavy metal from water by emphasizing the adsorption capability, effect of pH, isotherm and kinetic model, removal efficiency and the contact of time of every adsorbent.

**Keywords:** water pollution; heavy metals; adsorption; adsorbents

## 1. Introduction

Globally, environmental pollution has been the main cause of many deaths and illnesses [1]. Its consequences have been intensely felt by all mankind and its surrounding environment since the beginning of civilization, either consciously or not [2]. At first, pollution was not seen as a major issue because of the wide abundance of space for people to live in [3]. Eventually, it became a global concern and a serious threat to human life due to the rapid industrialization and geometrical development, mostly in urban cities [4]. Until recent years, it was studied that pollution accounts for 16% of all deaths globally and 9 million deaths nationwide, with a comparison of three times the deaths as malaria, tuberculosis, and acquired immunodeficiency syndrome (AIDS) put together [5]. Environmental pollution is undeniably responsible for more than one in four deaths [1] and the Earth is continuously threatened with the increasing anthropogenic or man-made activities causing severe damage to the Earth.

Water occupies 71% of the Earth chemically, 95.6% of which is kept in oceans that are not readily available for human consumption, unless a complicated desalination procedure is conducted [6]. According to an estimation from the World Health Organization (WHO), access to treated drinking water supplies is still lacking for 1.1 billion people worldwide, while about 2.4 billion people do not have access to proper sanitation [7]. In industries such as pharmaceuticals, food, electronics, etc.; clean water also serves as an essential feedstock. It should be acknowledged that accessibility to safe and accessible water is considered one of the most significant humanitarian priorities but continues to be a problem due to the rapid anthropogenic activities and population growth that also contribute to the release of pollutants, especially heavy metals particular to water bodies in particular [8,9].

Heavy metals have been one of the major contributing sources to water pollution throughout the decades. Water supplies may have naturally occurring ores rich in harmful

metals, which leach into water causing pollution. These ores are associated with occurrences of high arsenic and lead contamination [10]. One research has studied geological structures of various water source areas, which led to the findings of different heavy metals concentration in particular regions that can cause water pollution [11]. These heavy metals will not go through a decay process to non-toxic form [12], and therefore it creates an imbalance condition between the aquatic fauna and flora which later affects the human's health [13]. Adverse effects of heavy metals toward humans include stomach aches, vomiting, diarrhea, typhoid, cancer, hormonal imbalance, reproductive failure and serious damage to the liver and kidneys [14].

There are several common methods used in treating heavy metals from polluted water, such as phytoremediation, ion exchange, electrolysis, precipitation, ultrafiltration, coagulation, flocculation, reverse osmosis membrane and adsorption [15]. Therefore, this study will discuss how heavy metals affect the water bodies by reviewing several water treatments for environmental sustainability. In addition, a variety of adsorbents have been studied and utilized in the removal of heavy metals from water, along with their mechanisms in many research and review studies. The type of adsorbents can be classified into four categories: polymeric adsorbents, natural-based adsorbents, industrial by-product adsorbents and carbon nanomaterial-based. Hence, the summary of various recently studied adsorbents based on the adsorption capacity, the ideal pH and the suitable isotherm and kinetic model that fit the equation is reviewed.

## 2. Heavy Metals Pollution

Although there is no precise definition of what a heavy metal is, it has been described by literature as a naturally occurring element with a high atomic weight and high density equal to 5 $g/cm^3$ or more, which is at least five times greater than that of water [9]. Manganese, vanadium, chromium, iron, cobalt, nickel, copper, zinc, arsenic, molybdenum, silver, cadmium, lead, and mercury are several of the common heavy metals [10]. Heavy metals infuse into the environment through several activities such as industrialization, agricultural field, wastewater plant, runoff, metallurgical processes, mining and much more [12]. The toxicity of these metals will then be exposed to humans and animals via respiratory system, food consumption or direct ingestion of water containing heavy metal [9]. For various biochemical processes, there are heavy metals such as arsenic, cobalt, copper, iron, manganese, vanadium, and zinc that are considered essential elements since the body requires them in trace amounts; nonetheless, high concentrations of metals such as lead, mercury and cadmium without a doubt pose a significant threat or risk toward one's body [16].

### 2.1. Sources, Toxicity and Risk of Heavy Metals

Generally, the main sources of heavy metals in water can be classified into two categories, which are natural occurrence and man-made or anthropogenic activities [9]. Natural processes produce numbers of metal ions from geographic phenomena such as volcanic eruptions, rock weathering, leaching into rivers, lakes, and oceans due to water action [17]. It can also be explained that these heavy metals have been present naturally on the surface of the Earth since the beginning of Earth formation. The remarkable rise in the use of heavy metals has resulted in increasing heavy metals in the terrestrial and marine environments which has turned to become a major environmental challenge [18].

Conversely, heavy metal pollution has also emerged significantly due to anthropogenic activities, which is studied to be the prime cause of pollution. Mostly, such anthropogenic activities that contributes to increment of heavy metals include runoff, operation to mine and metal smelting, vehicles and roadworks, foundries and other industries that are metal based, coal combustion metal leaching from various sources, such as landfills, dump sites, excretion, manure from livestock, farm, and household [12]. In wind-blown dust, metals are mainly released from industrial areas. The secondary cause of heavy metal contamination also comes from the use of heavy metals in the agricultural sector, such as

the pesticides, insecticides, and fertilizers usage. It has become concerning since certain high concentrations of heavy metals have been observed to be threatening [9]. The heavy metals from natural occurrence or anthropogenic activities will distribute by diffusing into the air, immersing in soils or water, eventually reaching out to humans and animals as shown in Figure 1. Generally, metals deposited into the environment through atmospheric deposition, erosion of geological or anthropogenic activities.

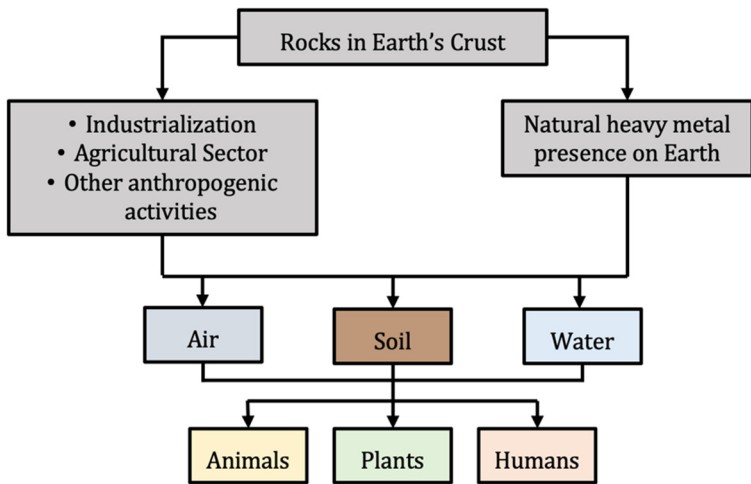

**Figure 1.** The heavy metal sources and their pathway into the environment and humans.

Table 1 represents the possible sources that might contribute to the increment of certain heavy metal ions in the water bodies, with respective harmful effect to humans and the permissible limit for each metal in order to be classified as safe water [16].

**Table 1.** Several heavy metal ions, with its respective possible source, health hazard and WHO (World Health Organization) permissible limit for water drinking. Adopted from reference [16] (copyright reserved Elsevier, 2020).

| Metal Ions | Sources | Harmful Effect | Allowed Limit (ppm) * |
|---|---|---|---|
| As(V) | Volcanic activity, industries, paints, drugs, dyes and textile, agriculture, smelting, mining | Severe arsenicosis, pigmentation problems, nausea, skin, and kidney cancer | 0.01 |
| Mo(II) | Industrialization, pesticides, catalysts, alloys, non-corrosive agents | Mineral imbalance, increment of serum ceruloplasmin, urinary copper excretion, gout-like symptoms | 0.07 |
| Zn(II) | Steel production plant, coal fire stations, galvanized metal pipes | Fever, vomit, nausea, cramp in stomach, diarrhea | 3.00 |
| Mn(II) | Mining, dumping sites, agriculture, fertilizers, soil | Nerve system failure, mutagenic and hepatic encephalopathy | 0.10 |
| Cd(II) | Electroplating plant, metal smelting, paints, batteries production, fertilizers, alloy industry | malfunction of renal, pulmonary troubles, bone cancer, high blood pressure, Itai-Itai disease, bone abnormalities | 0.01 |
| Co(II) | Metallurgy, mining, electroplating industry, paint manufacturing, nuclear power factory, tanning | Skeletal defects, diarrhea, hypotension pulmonary, paralyzed | 1.30 ** |
| Ni(II) | Nickel plating, alloys, production of batteries | Carcinogenic, losses of hair, skin toxicity | 0.10 ** |

| Metal Ions | Sources | Harmful Effect | Allowed Limit (ppm) * |
|---|---|---|---|
| Cu(II) | Battery manufacturing, plumbing corrosion | Headache, depression, low IQ | 1.30 |
| Pb(II) | Plumbing fixture, cable cover, ceramic, batteries, paints, wielding, extraction of lead, glass | Liver failure, neurological system damage, gastrointestinal tract impairment, high blood pressure, infertility, arthralgia | 0.05 |
| Hg(II) | Volcanic activity, mining operation, tanning, electroplating industries | Minamata disease, cancer | 0.002 |

Remarks: * The permissible limit for drinking water consumed are as stated by World Health Organization (WHO), except for Cobalt and Nickel statistics. ** The permitted Cobalt and Nickel limit for safe drinking water presented are referred from United State Environmental Pollution Agency (USEPA).

## 2.2. Heavy Metals Removal from Water

Natural occurrence and anthropogenic activities have been the main cause for the drastically rising concentration of heavy metals in the environment which caused severe health issues and permanent disabilities on living things when exposed [8]. Heavy metals that exceed its permissible limit are considered carcinogenic and lethal toward the environment as the effects will prolong until it can be noticeable in humans, animals, and plants, including at low intensity [10]. Therefore, throughout the decades, researchers and scientists have been developing numerous technological methods of remediation in order to prevent the surge of heavy metals contamination. Since heavy metals pollution is concerning, studies regarding elimination and reduction of heavy metals concentration have always sparked the interest of researchers in related environmental fields [8]. Thus, the environmental assessment for water pollution is summarized in Table 2 that leads to the creation of an ideal remediation method [15,16,19].

**Table 2.** The environmental assessment of water pollution.

| Assessment | Water Pollution | References |
|---|---|---|
| Source of Pollution | Industrialization, solid and liquid waste discharge, septic tanks system, eutrophication, mining, bio-solids, acidification, agricultural field, landfills, oil, and gas salt-water pits, neglected well sites, brine disturbance, combustion, impoundment of water, hydrocarbons. | [16] |
| Type of Pollutant | Heavy metals, industrial wastes, WWTPs, brewery, milk manufacturing plants, suspended solids, pathogens, Pharmaceutical and Personal Cosmetic Care (PPCPs), Persistent Organic Pollutants (POPs), toxic and nontoxic substances, pathogenic parasites, bio-concentrated metals, crop wastes, fertilizers, pesticides, plastics, shipwrecks, paper mills. | [19] |
| Risks | Cholera, kidney and heart dysfunction, terrible blood circulation, nauseous, nervous system problems, aquatic ecosystem impairment, death, diarrhea, typhoid. | [16] |
| Remediation Method | Source prevention actions, nutrients monitoring and control, disinfection, thermal treatment, bioremediation, phytoremediation, sewer, and proper septic tank usage, strengthened sustainable water quality and environment policies, water treatment method. | [15] |

## 2.3. Heavy Metals Removing Methods

Water pollution caused by heavy metal ions will eventually lead to high accumulation and concentration of the heavy metals that can severely affect public health. Therefore, in advance to reduce heavy metals' content in water, proper effluent treatment is needed so that a supply of clean water can be distributed into the environment or the household.

Furthermore, an environmental restoration field was introduced with several conventional procedures that can accomplish the objective of eliminating heavy metals ions from the environment. The methods that can be used to remove heavy metals include electrolytic based extraction, chemical precipitation, evaporation, reverse osmosis, ion exchange, electrochemical, membrane process and adsorption [20]. In general, there are three main classification of remediation methods which are by physical, chemical, and biological procedure. Table 3 shows the list of general methods that are applicable for removal of heavy metals [15].

**Table 3.** Common method for heavy metals removal in water. Adopted from reference [15] (copyright reserved Elsevier, 2021).

| Classification | Methods |
|----------------|---------|
| Physical | Adsorption<br>Ion exchange<br>Nanofiltration<br>Reverse osmosis membrane<br>Solvent extraction<br>Ultrafiltration |
| Chemical | Electric flocculation<br>Electrodeposition<br>Electrodialysis<br>Electrolysis<br>Ferrite precipitation<br>Insoluble salt precipitation<br>Neutralization precipitation |
| Biological | Bio-flocculation<br>Bio-sorption<br>Phytoremediation<br>Bio-precipitation<br>Biotransformation |

Throughout the years, pollutants, especially heavy metals have been eliminated using various physical, biological, and chemical treatment procedures by researchers. Among these treatment processes, coagulation/flocculation-sedimentation (CFS), ion-exchange, adsorption, membrane filtration and microbial degradation are noteworthy [6,15]. Each method has its respective strengths and weaknesses in dealing with the certain metal ions. Therefore, Table 4 summarizes the benefits and limitations of various water treatment processes [20–23]. Amongst the known methods, adsorption is usually considered as one of the most favored methods for heavy metals removal since it is a relatively simple techniques, numerous adsorbents availability, high efficiency, simple operation, good reversibility, affordable cost, and the regenerative ability of adsorbents [6]. In order to support the goal of discovering the most efficient water treatment system, the adsorption process fits the criteria to optimize the removal of heavy metals. Finding the perfect selection of a component termed as adsorbent is the key to any adsorption process. The basic characteristics of a good adsorbent should pose broad adsorption power, rapid rate of adsorption and easy separation and recovery.

**Table 4.** Summary of conventional wastewater treatment processes. Adopted from reference [22] (copyright reserved Elsevier, 2019).

| Method | Benefits | Limitations | Reference |
|---|---|---|---|
| Adsorption | A wide range of adsorbent, excellent adsorption capacity, simple and low cost. | Difficult regeneration of adsorbent and sludge, different adsorption capacity for different type of adsorbent. | [6] |
| Microbial degradation | Short process time, economically safe and produce non-hazardous product. | Toxic metal hinders microbial activity and possibility of clogging of pumping and injection wells. | [15] |
| Membrane filtration | Higher removal efficiency, No pollution loads, Removal of different contaminants. | Capital and running cost are high, Operation and Maintenance requirement cost, Toxic waste as product, Membrane fouling. | [20] |
| Ion exchange | High metal recovery, fewer sludge volume and limited pH tolerance. | Costly and high on maintenance | [21] |
| Coagulation/Flocculation | Quick process, inexpensive, straightforward process and coagulating agents are easily accessible. | Produce waste, low efficiency removal and required extra process such as sedimentation and filtration | [23] |

## 3. Remediation by Adsorption

The adsorption method has attracted the most interest to supply clean water in ancient cultures and is still commonly used in today's world because of its design and operation flexibility, low cost, simple and convenient to operate while still giving an excellent treatment efficiency [24]. Adsorption in general is a separation process that involves the removal of a substance (pollutants) from one phase (liquid or gas) followed by its accumulation at the other surface (adsorbent). In industries such as paper, dyes, textile, cosmetics and others, adsorption has been used to extract harmful organic such as endocrine disruptors, pharmaceutical ions and organic or inorganic contaminants from water [12]. Table 5 shows several adsorption mechanisms [25–27] that used adsorbents to remove targeted adsorbates.

**Table 5.** Several common adsorption mechanisms of adsorbent toward adsorbate. Adopted from reference [25] (copyright reserved Elsevier, 2020).

| Adsorption Mechanism | Mechanism Description and Illustration | References |
|---|---|---|
| Physical adsorption | Interaction between adsorbate and adsorbent surface through weak bonds, for instance weak van der Waals forces, hydrogen bonding or hydrophilic interactions.  | [25] |

<div align="center">

**Table 5.** *Cont.*

</div>

| | | |
|---|---|---|
| Electrostatic interaction | Attraction between the ion and the surface of the opposite charge, e.g., positive charged ions with a negative surface of an adsorbent.  | [26] |
| Ion exchange | A process involving the interchange of adsorbent and adsorbate with the matching ions charge.  | [21] |
| Surface complexation | A process where direct bond between adsorbate and adsorbent surface occurs at the inner sphere complex, while the outer sphere complex interacts with adsorbent via electrostatic interaction while retaining the hydration sphere.  | [27] |
| Precipitation | Solid formation in solution or on a surface when the adsorbate ions interact on the adsorbent surface with a surface functional group due to a pH change.  | [26] |

Zhu et al. [28] has focused on experiments that use biochar for heavy metal adsorption from cattle manure (CM) utilization since it was found that CM had better adsorption characteristics than rice husks, indicating that CM has an excellent adsorption capability. Cattle manure biochar (CMB) is highly efficient for heavy metals treatments and represents a new kind of cattle manure resource utilization. The utilization of biochar can be seen through all sorts of adsorption mechanisms as mentioned in Table 5. To fully grasp the

difference of each mechanism, an overview of how each mechanism is applied using biochar toward Methylene Blue (MB) is shown in Figure 2 [28].

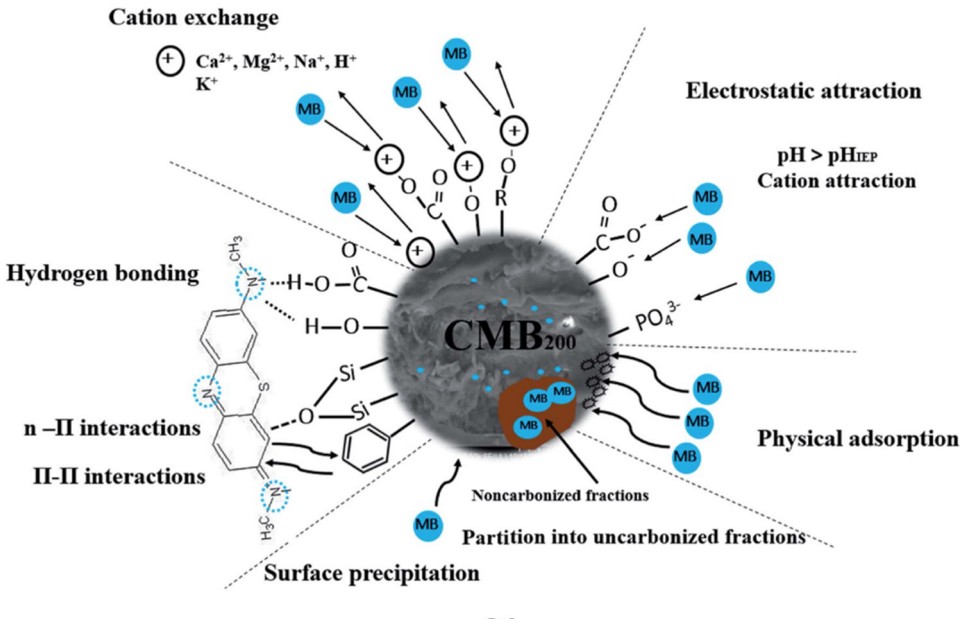

**Figure 2.** Interaction mechanisms involving adsorption in the biochar-MB system. Reproduced with permission from [28]. (Copyright reserved Royal Society of Chemistry, 2018).

### 3.1. Overview of Adsorption Phenomenon

The word adsorption was initially made known in 1881 by Heinrich Kayser, a German physicist at the time [29], and the term is currently used today. Adsorption is a surface phenomenon or a separation process wherein certain substances are removed from the fluid phases, known as gases or liquid, and it usually includes gas, liquid or solid molecules, atoms or ions attached to the surface, in a dissolved state. To simply describe, the ions or molecules are isolated via adsorption from aqueous solution onto surfaces of the solids (adsorbent) [24]. The process of adsorption is not to be mistaken with absorption as absorption conversely is a bulk phenomenon related to the uniform penetration and dispersal of one item into another.

Adsorption is studied to be reversible; thus, the reversed reaction of adsorption is known as desorption, where it releases the adsorbate [30]. Desorption is necessary for a material to serve as a good catalyst so that the products produced on the surface separate (desorbed) after the reaction to provide free surfaces for other reactant molecules to repeat the process again [30]. This is important in order to ensure that the process will continue to undergo adsorption within the free spaces. The rate of change in the adsorbate can also be written as a difference between adsorption and desorption [31]. The general mechanism of adsorption–desorption process with its terminologies is illustrated in Figure 3.

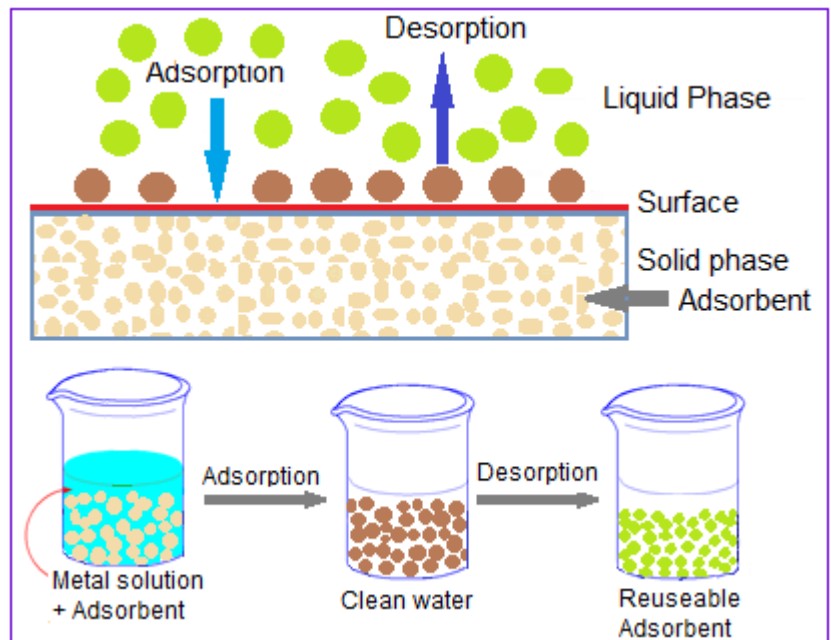

**Figure 3.** The mechanism of metal ions adsorption–desorption process in water.

### 3.1.1. Classification of Adsorption

For the adsorption process, it can be classified according to two categories of adsorption, which is chemical adsorption (chemisorption) and physical adsorption (physisorption) [24,32]. There will be cases where bonds of both types may be occurring at a time [31]. In chemisorption, adsorption occurs only in the area where chemical bonding between adsorbent and adsorbate such as ionic and covalent bonds are operating since chemisorption is highly specific and can only be achieved by the possibility of chemical bonding formation. The chemical bonding between both adsorbent and adsorbate leads to the enthalpy of the chemisorption to be within 200 to 400 kJ/mol. Chemisorption is strongly related to the surface area and temperature, as it will affect the adsorption efficiency [24].

Conversely, physisorption usually occurs in weak interactions such as weak van der Waals' forces. It is an exothermic process, where the enthalpy of the adsorption is lower, between 20 to 40 kJ/mol [24]. Unlike chemisorption, physisorption is not specific as the adsorbate can be adsorbed in all solids to the same extent. The overall physical adsorption is also affected by surface area and temperature variables, where it favors larger surface area but lower temperature. Table 6 compares the properties of chemisorption and physisorption [31] and Figure 4 shows the pathway before chemisorption and physisorption process can occur. To create an effective adsorbent, we need to identify the type of adsorbent to ensure that the proposed adsorbent has high selectivity and affinity toward the adsorbate. The diffusion potential is where the motion of adsorbate molecules at the adsorbent surface occurs. In physisorption, one of the diffusion transport steps acts as a rate-limiting step, while in the case of chemisorption, the adsorption step acts as the rate-limiting step [33]. After every aspect is fulfilled only then adsorption can occur, either physically or chemically.

**Table 6.** Comparison between chemisorption and physisorption.

| Description | Chemisorption | Physisorption |
|---|---|---|
| Activation energy | Required | Not required |
| Temperature | High | Low |
| Enthalpy | 200–400 kJ/mol | 5–50 kJ/mol |
| Common adsorption formation | Unimolecular layer | Multimolecular layer |

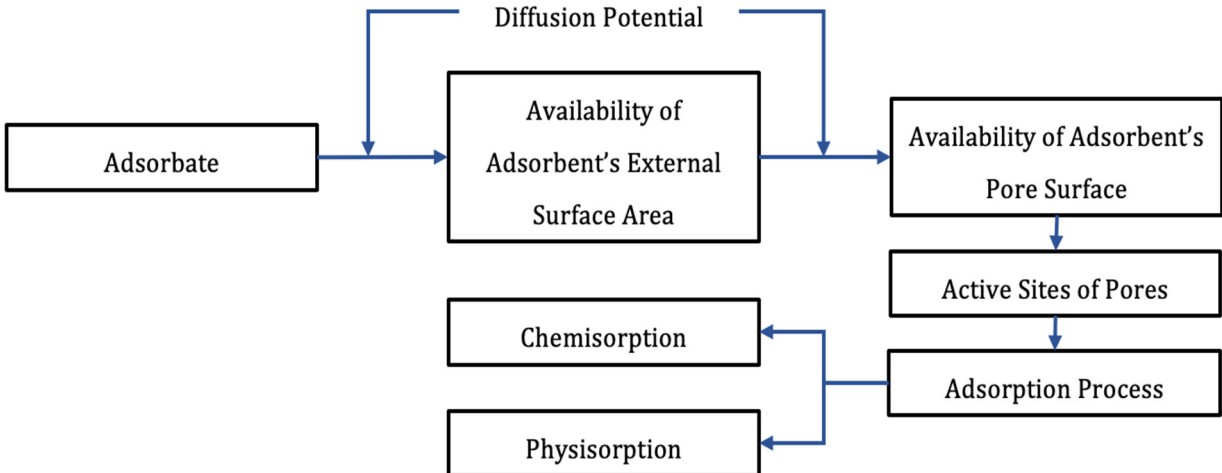

**Figure 4.** The adsorption pathway that leads to chemisorption or physisorption. Reproduced with permission from [34] (copyright reserved Elsevier, 2018).

### 3.1.2. Factors Affecting Adsorption

The performance of the adsorption process is strongly related to several of these important parameters, which are pH, the pH at the potential of zero-point charge ($pH_{zpc}$), adsorbent dosage, temperature, pressure, surface area and coexisting ions [35]. First and foremost, pH is one of the most significant parameters that influences the effectiveness of pollutants or adsorbate removal by adsorbents. The ionization, speciation of adsorbate in solution and the surface nature of the adsorbent are heavily influenced by pH due to the interaction of hydrogen ions ($H^+$) and hydroxide ions ($OH^-$) at the active site of adsorbent.

The next factor affecting the adsorption process is the pH at the potential of zero-point charge, also known as $pH_{zpc}$ where adsorbent surface charge plays a key role in the batch processes and helps in understanding the mechanism of sorption [33]. The $pH_{zpc}$ is the pH at which the surface charge of the adsorbent is equal to zero. The theory of it will allow the hypothesizing that material surface is positive below the point of zero charges and will allow negatively charged contaminants or pollutants to be adsorbed. The electrostatic attraction will increase the rate of adsorption among oppositely charged adsorbent species thus increasing the process efficiency. Two common techniques in determining the $pH_{zpc}$ are mass technique and immersion technique [33]. The next affecting factor is the adsorbent dosage. Since adsorbent requires an active site for the adsorption to occur, therefore having a higher dosage of adsorbent will increase the active adsorption sites will be more effective in removing the contaminants or pollutants [35]. Despite that a higher dosage will create more active sites, the increasing dosage will also indirectly reduce the total uptake of pollutants ($q_e$) per unit mass of an adsorbent since the presence of unsaturated sites are also present in the process [35].

The following key factor is temperature, where specifically in the batch process, the temperature will alter the adsorbent characteristics, adsorbate stability and the adsorbate-adsorbent interaction [33]. The viscosity of the solution eventually decreases as temperature increases, which helps to move contaminants from the bulk solution to the surface of the substance. Thermodynamic studies help to estimate the characteristics of the batch adsorption process, whether it is exothermic or endothermic, spontaneous or random type and also suggest the favorable temperature batch adsorption process. The negative values of Gibbs free energy ($\Delta G_0$) are responsible for the spontaneity of adsorption and the enthalpy ($\Delta H_0$) implies the principle of the phase, whether it is an exothermic or endothermic process. Le Chatelier's principle can be applied to explain the magnitude of adsorption for both chemisorption and physisorption as shown in Figure 5 [33].

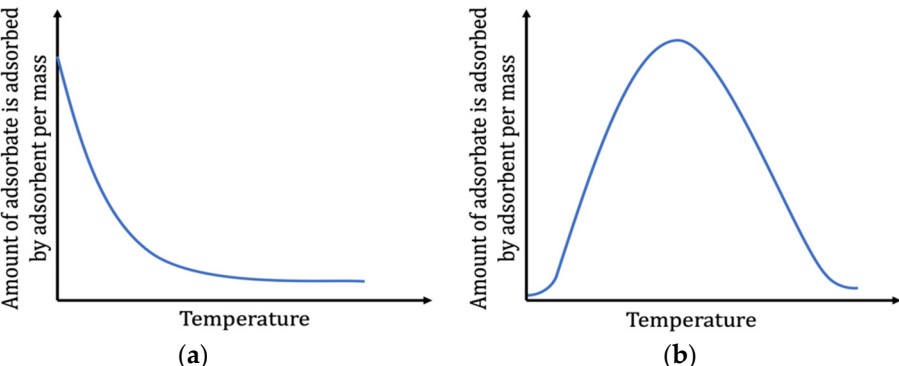

**Figure 5.** Effect of temperature on (**a**) physisorption and (**b**) chemisorption.

From the aspect of adsorption isotherm that involves the principle of pressure, an increasing pressure will also intensify the adsorption process to a higher level until the adsorbent becomes saturated [33]. However, when the adsorption has reached the equilibrium level, the pressure will no longer play an important role in making the adsorption process more effective, no matter how much pressure is being applied on the process. Therefore, the general graph with pressure as a function is shown in Figure 6.

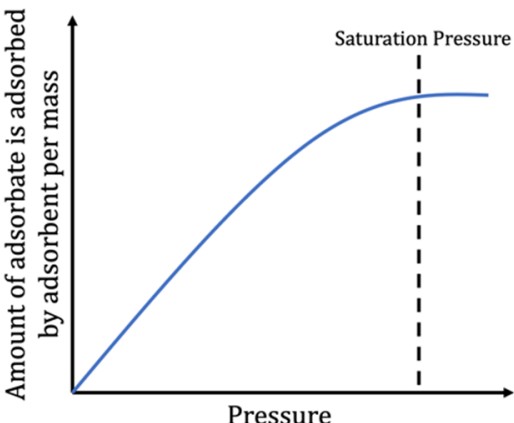

**Figure 6.** Graph of adsorption capacity against pressure in batch adsorption.

Surface area also plays an important role in affecting the overall adsorption process. Since adsorption is categorized as a surface phenomenon, it is strongly related with the specific surface, where adsorption by smaller particles is relatively better compared to the larger particles. This can be achieved by breaking up larger particles into smaller ones that open certain small scales, hence increasing their surface area for greater adsorption [33]. Lastly, the presence of co-existing ions will also affect the uptake capacity of the adsorbent as increasing active sites will also increase the competition of the material surface within themselves. Since water typically consists of various component types, the mixture of compounds sometimes increases the mechanism of adsorption and at certain times interferes with another one. Competition between adsorbates in mixed solutions depends on the molecular size of the adsorbents, the concentration of the solutes and their relative affinity [33]. Table 7 summarizes the roles of each parameter in affecting the overall adsorption process.

**Table 7.** Summary of factors that are affecting the overall adsorption's process.

| Factor Affecting Adsorption | Effect on Adsorption Process |
|---|---|
| pH | The presence of hydrogen ions ($H^+$) and hydroxide ions ($OH^-$) will react with the activated site of the adsorbent depending on the pH value. |
| pH at the potential of zero-point charge ($pH_{zpc}$) | Assuming that the adsorbent's surface is positive below the point of zero charges, it will allow negatively charged contaminants or pollutants to be adsorbed in batch processes. |
| Adsorbent dosage | Increase the active adsorption sites will be more effective in removing the contaminants or pollutants, although too much dosage will reduce the total uptake of pollutants ($q_e$) |
| Temperature | Higher temperature reduces the viscosity, which helps the mobility of contaminants from the bulk solution to the surface of the substance. |
| Pressure | Intensify the adsorption process up to a higher level until the process reaches equilibrium. |
| Surface Area | Smaller particles have bigger surface area compared to the larger particles, allowing greater adsorption to occur. |
| Co-existing ions | Lesser co-existing ions in the solution will have a better adsorption process. |

*3.2. Chelating Resins*

It is studied that adsorption is a good remediation method due to its effectiveness, simple operation method, broad range of sorbent availability and reusability that help the environmental protection [6]. Additionally, at the same time, functionalization of chelating resins into various types of adsorbents are widely experimented for the metal ions removal since these resins have high selectivity and capacity to adsorb which enhances adsorption process while still being a low-cost and simple method [36,37]. The main chain (parent) of chelating resins are crosslinked polymers, usually grafted with special functional groups for certain unique purposes. The nature of these polymers is associated with the presence of phosphate, oxygen, nitrogen, and sulfur chelating groups with absorbability features. Chelating resins prove a greater adsorption selectivity compared to conventional small molecule chelating agents due to the good synergy, electrostatic, stereo, and polymeric effect such as concentration and dilution of chelating resins' functional groups [38].

In addition, the molecular framework and the three-dimensional molecular structure of the resin body are insoluble in solvents that are organic such as alkali, acid, and water, thus making the separation easier. It also has rapid adsorption toward metal ions, high selectivity, high capture capacity and resource recovery [39]. In late 1970s, the possibility for a resin to be applied to any complexation process in analytical chemistry was studied, conducted, and concluded that the main obstacle of this study is to prevent any ligand groups such as thiol or primary amino groups from losing during the coupling reaction [40]. A study has successfully proved analytical applications of resins containing amide and polyamine functional groups to make sure it is selective for one particular metal ion or one from which several metal ions until it can be retained and eluted [41]. Currently, a broad range of chelating resins such as iminoacetate, amino phosphonate, hydroxamic acid and amidoxime has been utilized to efficiently remove the metal ions [42,43]. The amidoxime and hydroxamic acid groups have an excellent chelation reaction against many rare earth elements and heavy metal which explains why chelating adsorbent carrying amidoxime groups and hydroxamic acid groups attract interest in the recovery, enhancement and removal of rare earth and heavy metal elements [44,45].

Application of Chelating Resin in Removing Heavy Metals

The existing chelating resins with its dual functionality can access a variety of applications due to their inherent properties of air-water interfaces adsorption capability, adsorb ability and high affinity of complexation for metal ions of higher valency. Chelating resins is widely used to separate the certain metal ion from its aqueous solution, water bodies or from the soil efficiently, either to remove or to retrieve the metals for other uses [39]. The chelating resins that contribute to several applications that are summarized in Table 8, with their respective purpose, along with the highlighted methods and chelating resins [39–42].

**Table 8.** Highlighted applications of chelating resins with their respective method and chelating resins examples. Adopted from reference [32] (copyright reserved Elsevier, 2019).

| Application | Purpose | Method | Chelating Resins | Reference |
|---|---|---|---|---|
| Pollution Control | To remove any metal ions contaminant that degrades the environment.  | Adsorption | Silica supported amidoxime ligand | [46] |
| Pre-concentration system | To enhance the detection of trace metal ions.  | Flow Injection Analysis | PSDVB-PAN | [47] |
| Removal of metal ions | To discharge non-surface-active ions from aqueous solutions by adding a surfactant followed by air flotation and assortment of formed foam.  | Ion Flotation | Surface-Active Derivative of $C_{12}$-DTPA | [48] |
| Biology processes | Act as an inhibitor for enzymes that contains metals, known as metalloenzymes.  | Extraction | 2-alkylmalonic acid amphiphile | [49] |

Abbreviations: amphoteric 2-dodecyldiethylenetriamine penta-acetic acid ($C_{12}$-DPTA), polystyrene divinylbenzene resin functionalized by 1-(2-pyridylazo) 2-naphtol) (PSDVB-PAN).

### 3.3. Removal of Heavy Metals by Various Adsorbents

In the past several decades, adsorbents have been developed from various materials for the removal of pollutants, in this context heavy metals from water and wastewater. The adsorbents can be generally categorized into two main classes, organic and inorganic, where the classification refers to the base material used for the fabrication of adsorbents. Figure 7 illustrates the categorization of adsorbents, and Table 9 lists several commonly used or studied examples for each categorization [34,50].

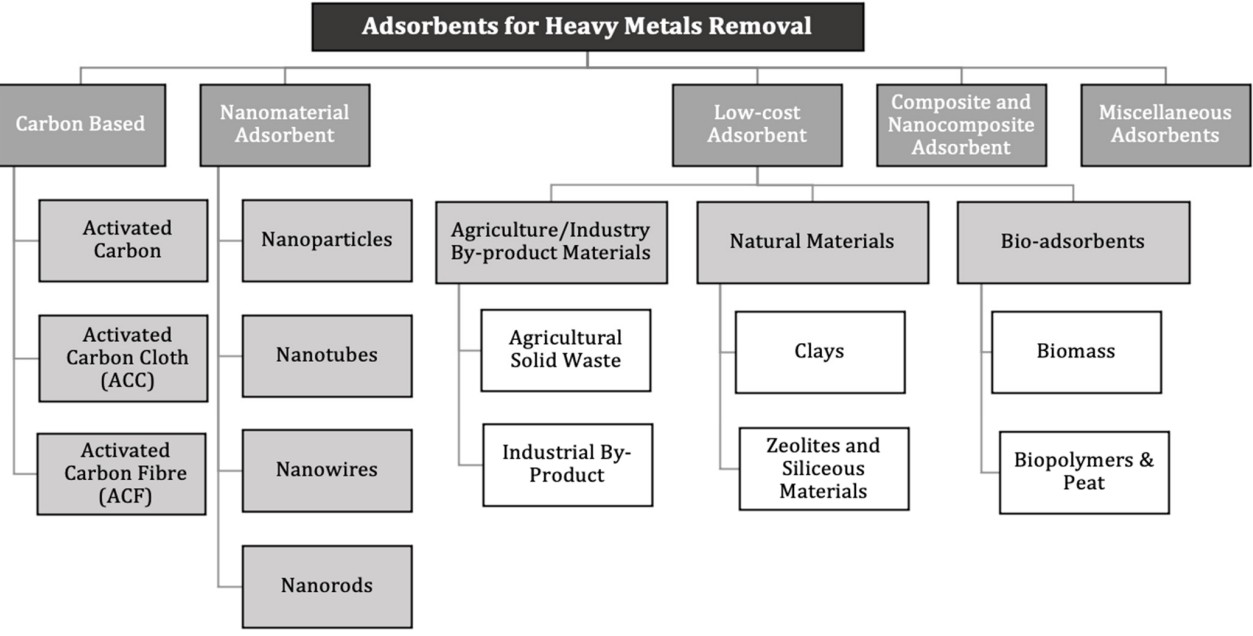

**Figure 7.** The classification of adsorbents used for remediation methods. Adopted from reference [34] (copyright reserved Elsevier, 2018).

**Table 9.** Lists of adsorbent classification. Adopted from reference [34] (copyright reserved Elsevier, 2018). Adopted from reference [50] (copyright reserved Elsevier, 2021).

| Adsorbent Classification | Examples |
|---|---|
| Polymeric based | • Poly(acrylic acid) (PAA)<br>• Poly(amidoxime)<br>• Poly(hydroxamic acid)<br>• Polycyclic Aromatic<br>• Cellulose Nanofiber<br>• Polymeric Dextran<br>• Bifunctional Resins |
| Industrial By-product | • Metal Hydroxide Sludge<br>• Fly Ash<br>• Red Mud<br>• Bio-solids<br>• Waste Slurry |
| Natural Minerals based | • Silica<br>• Zeolite<br>• Alumina<br>• Kaolinite<br>• Bentonite<br>• Montmorillonite<br>• Layered double Hydroxide (LDH)<br>• Clays<br>• Chitosan |

**Table 9.** *Cont.*

| Adsorbent Classification | Examples |
|---|---|
| Carbon Nanomaterial based | • Graphene Oxide (GO)<br>• Carbon Nanofiber<br>• Carbon Nanotubes (CNTs) |

### 3.3.1. Polymeric Adsorbent

Polymeric adsorbents have become a major adsorption research focus on recent years. This is due to their appealing characteristics such as physical and chemical stability and rigidity, along with the heat and environmental resistance. Furthermore, due to their flexibility in synthesis and functionalization, polymeric adsorbents have a high adsorption capacity for eliminating a wide range of contaminants with varying properties, especially heavy metals [50]. Incorporating two distinct polymers or permanently scattering inorganic nanoparticles within polymer supports has resulted in polymer–polymer hybrids and polymer–inorganic hybrids. Across the combination of two counterparts at a nanoscale range, the formed composites not only maintain the intrinsic features, but also typically demonstrate higher processability, more stable stability and fascinating advances produced by the interaction of nanoparticles–matrix [51]. Surface functional groups such as amino and hydroxyl groups that mostly exist on polymers, have a strong affinity for heavy metal ions [24]. Therefore, Table 10 represents various polymer-based adsorbents with all related heavy metal adsorption capacities, with the optimum pH, preferred kinetic and isotherm model [52–57].

**Table 10.** Heavy metal adsorption capacities of various polymer-based adsorbents.

| Adsorbent | Target Metal | Optimum pH | Adsorption Capacity (mg g$^{-1}$) | Kinetics Model | Isotherm Model * | Reference |
|---|---|---|---|---|---|---|
| Poly(amidoxime)-palm cellulose | Cu | 6 | 260.0 | Pseudo-first order | F | [52] |
| | Fe | 6 | 210.0 | Pseudo-first order | F | |
| | Co | 6 | 168.0 | Pseudo-first order | F | |
| | Ni | 6 | 172.0 | Pseudo-first order | F | |
| | Pb | 6 | 272.0 | Pseudo-first order | F | |
| Poly(amidoxime)-jute cellulose | Cu | 6 | 310.0 | Pseudo-second order | F | [53] |
| | Co | 6 | 295.0 | Pseudo-second order | F | |
| | Cr | 6 | 227.0 | Pseudo-second order | F | |
| | Ni | 6 | 175.0 | Pseudo-second order | F | |
| Poly(amidoxime)-waste cellulose | Cu | 6 | 298.4 | Pseudo second order | L | [54] |
| | Co | 6 | 289.6 | Pseudo-second order | L | |
| | Cr | 6 | 217.0 | Pseudo-second order | L | |
| | Ni | 6 | 168.7 | Pseudo-second order | L | |
| Poly(hydroxamic acid)-kenaf cellulose | Cu | 6 | 305.3 | Pseudo-second order | L | [55] |
| | Fe | 6 | 275.6 | Pseudo-second order | L | |
| | Mn | 6 | 258.5 | Pseudo-second order | L | |
| | Co | 6 | 256.6 | Pseudo-second order | L | |
| | Cr | 6 | 254.3 | Pseudo-second order | L | |
| | Ni | 6 | 198.5 | Pseudo-second order | L | |
| | Zn | 6 | 190.1 | Pseudo-second order | L | |
| Poly(hydroxamic acid)-palm cellulose | Cu | 6 | 325.0 | Pseudo-first order | F | [56] |
| | Fe | 6 | 220.0 | Pseudo-first order | F | |
| | Pb | 6 | 300.0 | Pseudo-first order | F | |

<div align="center">

**Table 10.** *Cont.*

</div>

| Adsorbent | Target Metal | Optimum pH | Adsorption Capacity (mg g$^{-1}$) | Kinetics Model | Isotherm Model * | Reference |
|---|---|---|---|---|---|---|
| Poly(hydroxamic acid)-jute cellulose | Cu | 6 | 352.0 | Pseudo-first order | F | [53] |
| | Co | 6 | 318.0 | Pseudo-first order | F | |
| | Cr | 6 | 230.0 | Pseudo-first order | F | |
| | Ni | 6 | 188.0 | Pseudo-first order | F | |
| Poly(hydroxamic acid)-waste fiber | Cu | 6 | 346.7 | Pseudo-second order | L | [57] |
| | Co | 6 | 315.0 | Pseudo-second order | L | |
| | Cr | 6 | 227.6 | Pseudo-second order | L | |
| | Ni | 6 | 181.4 | Pseudo-second order | L | |

Remarks: * Freundlich isotherm (F) or Langmuir isotherm (L).

### 3.3.2. Industrial By-Product Adsorbent

Industrial wastes, also known as the by-products of industries, can be also considered as adsorbents for the heavy metals removal from water and wastewater. This is because these sources have the capability to adsorb the heavy metals from water. Most industrial by-products will not be used for any other purposes, except in the adsorption process [58]. Industrial by-product adsorbents are certainly beneficial in terms of the availability and economical. Therefore, these industrial wastes have been discovered to be effective adsorbents [58]. Adsorptive capacity of these wastes could be increased followed by slight processing and modification as well. Table 11 shows several studies that use industrial by-products as its adsorbent for the removal of heavy metals [59–61]. Recent studies have a better fit with the pseudo-second order kinetic model and Langmuir isotherm model. The alumina refining business produces red mud (RM), which is a solid by-product [59]. Another industrial by-product, which is fly ash, that is obtained from coal power plants is studied to be effective as an adsorbent in the heavy metal removal as well. However, there is recent research that retrieved fly ash and bottom ash produced by biomass-based thermal power plants for removal of heavy metals and the study is observed to have high affinity toward heavy metals too [60].

<div align="center">

**Table 11.** Heavy metal adsorption capacities of various industries by product-based adsorbent.

</div>

| Adsorbent | Target Metal | Optimum pH | Adsorption Capacity (mg g$^{-1}$) | Kinetics Model | Isotherm Model * | Reference |
|---|---|---|---|---|---|---|
| Red mud | Pb | 5 | 128.53 | Pseudo-second order | N/A | [59] |
| | Zn | 5 | 35.70 | Pseudo-second order | N/A | |
| Fly ash | Pb | 6 | 194.70 | Pseudo-second order | L | [60] |
| | Cu | 6 | 151.40 | Pseudo-second order | L | |
| | Cd | 6 | 143.10 | Pseudo-second order | L | |
| | Zn | 6 | 92.60 | Pseudo-second order | L | |
| Bottom ash | Pb | 5-6 | 53.20 | Pseudo-second order | L | [60] |
| | Cu | 5-6 | 32.40 | Pseudo-second order | L | |
| | Cd | 5-6 | 23.60 | Pseudo-second order | L | |
| | Zn | 5-6 | 15.80 | Pseudo-second order | L | |
| Biochar supported zero-valent iron nanocomposite | As | 4.1 | 124.5 | Both PFO and PSO | F and L | [61] |

Remarks: * Freundlich isotherm (F) or Langmuir isotherm (L), N/A is no data available.

### 3.3.3. Natural Mineral Based Adsorbent

Another source of adsorbents that has been attracting researchers' interest is natural mineral based. This is because most natural mineral based adsorbents are low-cost, abundant, making it easier to be retrieved, besides its excellent adsorption capability [34,62].

Although there are many existing natural minerals on this Earth: chitosan, silica, zeolite and clays including kaolinite, bentonite, montmorillonite are studied to be effective in heavy metal removal. In recent studies, chitosan and clays are frequently altered and modified with other adsorbents to improve its effectiveness. Using the co-condensation approach, a study by Yin et al. [63] focused on synthesizing a silica sorbent functionalized with amidoxime groups. With a maximum uranium extraction capacity of 3.36 to 3.94 mg g$^{-1}$, the functionalized silica sorbent is proved to be able to remove uranium from saline lake brine. Table 12 highlighted several recent studies that used natural mineral based adsorbent for the removal of heavy metals, by using the adsorption process with its respective targeted metals, kinetic and isotherm model [64–66].

**Table 12.** Heavy metal adsorption capacities of various natural mineral-based adsorbents.

| Adsorbent | Target Metal | Optimum pH | Adsorption Capacity (mg g$^{-1}$) | Kinetics Model | Isotherm Model * | Reference |
|---|---|---|---|---|---|---|
| Carboxymethyl chitosan–hemicellulose | Cu | 6 | 362.30 | Pseudo-second order | L | [64] |
| | Cr | 4 | 909.10 | Pseudo-second order | L | |
| | Hg | 4 | 333.30 | Pseudo-second order | L | |
| | Ni | 4 | 42.00 | Pseudo-second order | L | |
| | Cd | 4 | 28.20 | Pseudo-second order | L | |
| | Mn | 4 | 49.00 | Pseudo-second order | L | |
| Poly(amidoxime)-silica | Cu | 6 | 172.00 | Pseudo-first order | F | [46] |
| | Fe | 6 | 168.00 | Pseudo-first order | F | |
| Carboxylate functionalized-chitosan co-polymer | Pb | 6 | 127.91 | Pseudo-second order | L | [65] |
| | Cu | 6 | 123.50 | Pseudo-second order | L | |
| | Cd | 6 | 108.42 | Pseudo-second order | L | |
| | Zn | 6 | 92.27 | Pseudo-second order | L | |
| Synthetic NASO Zeolite (Na$_6$Al$_6$Si$_{10}$O$_{32}$.12H$_2$O) | Cd | 5 | 649.00 | Pseudo-second order | L | [66] |
| | Pb | 5 | 210.00 | Pseudo-second order | L | |

Remarks: * Freundlich isotherm (F) or Langmuir isotherm (L).

### 3.4. Heavy Metals Removal Efficiency by Several Common Functionalized Adsorbents

To ensure the efficiency of the adsorption process can be utilized to the maximum, a precise selection of adsorbent is the key factor. In general, the basic characteristics of a good adsorbent should exhibit broad adsorption capacity, rapid rate of adsorption, easy to separate or recover from the water [7], high porosity and small pore diameter, since higher surface area exposure leads to higher ability of adsorption [67]. Overall, numerous types of materials have been applied as adsorbent for many applications such as water treatment, catalysis, desiccants, indicators, and catalyst [67]. As mentioned from the previous subsection, certain commonly used adsorbents are activated carbon, cellulose, natural minerals, silica, biopolymers, and nanomaterials [15,50,68,69].

Adsorbents such as silica, zeolite and activated carbon are favorable since they are low-cost and can be retrieved from natural compounds, besides having excellent adsorption ability [15]. Activated carbon (AC) is the commonly used and popular adsorbent for the decontamination wastewater due to its high surface area and high tolerance for heavy metals and dye molecules. There are several types of activated carbon used for removal of pollutants from water, which are granular activated carbon (GAC), powdered activated carbon (PAC) and activated carbon cloth (ACC) [34]. Conversely, zeolites that take the form of aluminosilicates or other interrupted structures of zeolite-like materials, such as aluminophosphates, also could adsorb heavy metals [68]. Natural zeolites are rated as the cheapest alternative adsorbents among the current commercial adsorbent prices in comparison to other commercial-grade products, making zeolite a great alternative adsorbent other than its high porosity [68].

Lastly, silicas also have received extensive attention as promising sorbents and have opened a wide field of applications. This is due to how the silica can be functionalized

into silica gels with large specific area, strong mechanical characteristic, also good chemical and thermal stability [70] or mesoporous silicas such as Mobil Composition of Matter No. 48 (MCM-48) and No. 41 (MCM-41), hexagonal mesoporous silica (HMS) and Santa Barbara Amorphous-15 (SBA-15) that are considered superior in adsorption since it has wide surface area, regulated pore sizes and small pore-size distributions [71]. These functionalized silicas can be used efficiently as adsorbents to extract heavy metal ions and other harmful pollutants [72]. Despite that silica alone has good adsorption ability, it still has few downsides in adsorbing soft metals such as lead, tin, zinc, aluminum, thorium, copper, and bronze [73]. To fully distinguish each adsorbent, a pore size scheme of zeolites, mesoporous silica, and metal-organic framework (MOFs) is shown in Figure 8 [74], and a comparison study of the adsorption capability between activated carbon, zeolites and silica on several targeted contaminants from water is presented in Table 13 [75].

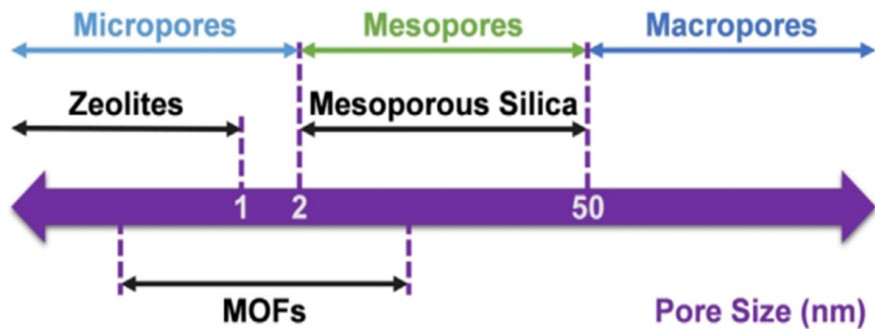

**Figure 8.** The pore size scheme for zeolites, mesoporous silica, and MOF. (Reproduced with permission from [74]. Copyrights reserved Elsevier, 2020).

Another alternative to evaluate the adsorption capacity of various adsorbents is by observing the removal percentage (%) of the heavy metals by the adsorbent. The percentage of adsorbate removal will describe the ability of the adsorbent to adsorb heavy metals. Table 14 highlights the removal efficiency of several commonly used adsorbents, along with the conditions used on respective adsorption processes [76–80]. Most functionalized adsorbents have an excellent adsorption ability toward targeted heavy metals (>90%), when enhanced with optimum pH condition, contact time and concentration. Certain adsorptions prefer an alkali condition although most adsorbed metals recorded the highest adsorption capacity in an acidic environment. However, most of the studies conducted were performed in a laboratory scale and the full potential, while being used in actual wastewater treatment remains uncertain [76–80].

**Table 13.** The concentration of pollutants before and after adsorption of silica, zeolite and activated carbon [75].

| Pollutant | Initial Pollutant Concentration (mg/L) | Final Concentration with Silica (mg/L) | Final Concentration with Zeolite (mg/L) | Final Concentration with Activated Carbon (mg/L) | Permissible Level in Drinking Water (mg/L) |
|---|---|---|---|---|---|
| Ammonium | 5.50 | 4.70 | 1.50 | 3.50 | 1.50 |
| Iron | 0.55 | 0.10 | 0.50 | 0.35 | 0.3 |
| Phosphate | 4.0 | 2.80 | 1.20 | 2.50 | N/A |
| COD | 200 | 70.0 | 180 | 21.0 | 0 |
| Turbidity | 100 * | 8.1 * | N/A | 9.7 * | 5 * |

Remarks: * is in NTU (nephelometric turbidity unit), N/A is no data available.

**Table 14.** The removal efficiency of several commonly used adsorbents toward heavy metals.

| Adsorbent | Target Metal | pH | Initial Metal Concentration (mg/L) | Contact Time (min) | Adsorption Capacity (mg/g) | Removal Percentage (%) | Reference |
|---|---|---|---|---|---|---|---|
| Activated carbon from African palm fruit | Cd | 8 | 1820.00 | 60 | N/A | 99.23 | [76] |
| | Cu | 3 | 1520.00 | 60 | N/A | 96.71 | |
| | Ni | 8 | 3240.00 | 60 | N/A | 95.34 | |
| | Pb | 3 | 2620.00 | 60 | N/A | 97.75 | |
| Magnetic graphene oxide | Pb | 5 | 60.00 | 25 | 200.00 | 89.61 | [77] |
| | Cr | 6 | 60.00 | 35 | 24.330 | 92.03 | |
| | Cu | 6 | 60.00 | 25 | 62.89 | 92.43 | |
| | Zn | 7 | 60.00 | 35 | 63.69 | 90.38 | |
| | Ni | 8 | 60.00 | 25 | 51.02 | 92.23 | |
| Silica oxide encapsulated natural zeolite | Pb | N/A | 10.00 | 30 | 186.00 | 99.30 | [78] |
| | Cu | N/A | 10.00 | 480 | 10.30 | 98.90 | |
| | Cd | N/A | 10.00 | 60 | 12.30 | 98.30 | |
| | Zn | N/A | 10.00 | 60 | 9.00 | 97.10 | |
| | Mn | N/A | 10.00 | 30 | 4.20 | 54.00 | |
| Plasma polymer functionalized silica | Cu | 5.5 | 15.00 | 60 | 25.00 | >96.70 | [79] |
| | Zn | 5.5 | 15.00 | 60 | 27.40 | >96.70 | |
| Polyaniline grafted cross-linked chitosan beads | Cd | 6 | 40.00 | 60 | 145.00 | 99.60 | [80] |
| | Pb | 5 | 40.00 | 60 | 114.00 | 99.30 | |

Remarks: N/A is no data available.

## 4. Conclusions

In conclusion, the heavy metal remediation method is an interesting field to study and explore. This review highlighted several water treatment methods that can potentially remove heavy metals from water effectively. The adsorption process is reviewed to be the preferrable remediation method since it offers numerous benefits in removing the heavy metals. The efficiency of the adsorption process is influenced by several parameters such as pH, adsorbent dosage, temperature, pressure, surface area and coexisting ions. The fabrication and functionalization of various adsorbents has been an emerging approach to ensure greater adsorption process. It can also be seen that most adsorbents fitted the Langmuir or Freundlich isotherm, indicating that they are either monolayer or multilayer adsorption. Most of these emerging adsorbents have excellent adsorption capacity when enhanced with the ideal pH, contact time and concentration. Although many efforts have been taken to remove the heavy metal ions from water, the upcoming challenge is to expand the application of the adsorbent from a laboratory scale to fit the industrial scale purpose. In order to fully prove the sustainability of various adsorbents in water treatment, a detail study for every aspect in the adsorption study should be covered in the future involving comparison between pH and $pH_{zpc}$, effect of contact time, regeneration of adsorbents, removal ratio of adsorption process, difference of concentration and considering having a broader isotherm model.

**Author Contributions:** Writing—original draft preparation, M.Z.A.Z.; writing—review and editing, M.L.R.; funding acquisition, M.S.S. All authors have read and agreed to the published version of the manuscript.

**Funding:** This research received no external funding.

**Institutional Review Board Statement:** Not applicable.

**Informed Consent Statement:** Not applicable.

**Data Availability Statement:** Not applicable.

**Acknowledgments:** The authors would like to express gratitude for the financial support by the UMS innovation grant (SGI0061-2018) for this paper.

**Conflicts of Interest:** The authors declared no conflict of interest.

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
