# Peer review of "Heavy Metals Removal from Water by Efficient Adsorbents"

_water, doi:10.3390/w13192659_

Round 1

Reviewer 1 Report

The manuscript presents review on the review of heavy metals removal from water by polymeric adsorbents. From the view of scientific research this paper is suitable for publication in Water. But there are still some obvious flaws.

1, The introduction part, the first paragraph is about the environmental pollution, second paragraph mentioned the “Various physical and chemical methods have been proposed to eliminate pollutants from the atmosphere, but most of the methods produce supplementary environmental issues and are costly.” While the third paragraph again back to the globalization. Then the fourth move to crisis of water resources. These indicated the introduction lacks logic.

  1. This manuscript mainly present the method of heavy metal removal, this should be present at great length. 2.1. Sources, Toxicity and Risk of Heavy Metals is of low necessity.
  2. 2.6.1 Classification of Adsorption. “chemisorption prefers high pressure conditions” ? Misleading.
  3. 2.6.1 Classification of Adsorption. Physical adsorption is generally accompanied by chemical adsorption.
  4. 2.6.2. Affecting Adsorption There are many important parameters should be considered such as functional groups, species of metal ions et al.
  5. 2.11. Porous Adsorbent Materials Granular materials are too ambiguous. There are lots of pranular materials such as porous organic polymers exhibited high specific surface area.
  6. Removal ratio with different materials should be discussed.
  7. 3. Conclusion : “ In conclusion, although the study of adsorbent with the presence of both amidoxime and hydroxamic acid ligand is very limited, it can be studied that silica can be successfully bind with both amidoxime and hydroxamic acid ligand to act as a promising adsorbent with better adsorption capability due to its porosity, structure order and high surface area.‘’ In this part prospect of removal materials should be present but not the limited of amidoxime or hydroxamic acid ligand.

Reviewer 2 Report

The authors reviewed and up-to-date information on polymer adsorbents in the treatment of heavy metal ions from water and wastewater.The subject has a significant importance regarding the actual water treatment performances. But few minor aspects has to be considered before accepting the paper:

  1. The abstract and conclusions emphasize special types of polymers (amidoxime, hydroxamic acid, acrylic,,,,,) but the paper consists in a large types of other adsorbents, not only polymers (for example, subchapters 2.9 and 2.11). All these adsorbents has efficiency for heavy metals, but for a clearly understanding of a subject either exclude these chapters where polymers are not the subject of adsorption or change the title.
  2. Some minor changes has to be made: line 281 (cut "an"); Table 6 (Illustration)
  3. Adsorption Phenomenon (2.6) is linked with Table 6. Maybe subchapter 2.4 is better to be integrated into 2.6 or viceversa.
  4. Subchapter 2.7 refers to pollutants (as general), but the paper is about heavy metals. These presented adsorbents are all specific for heavy metals? And all adsorbents are polymers?

Reviewer 3 Report

Review of manuscript: water-1358083

It is an interesting script, but it is not written properly. There are some points, which require revision and need to be clarified in the revised text. The points are described below.

  • General: You should reconstruct the tables of contents. For example, 2.3.1 Chelating Resins and 2.3.2 Application of chelating Resin were moved into “Remediation by a Adsorption”. 3. Remediation by a Adsorption, 3.1 Overview of Adsorption and desorption, 3.2 Adsorption Phenomenon, 3.3 Removal of Impurity by adsorbent, 3.3.1 Polymer Adsorbent, 3.3.2 Industrial By-product Adsorbent, 3.3.3 Natural Mineral Based Adsorbent, 3.3.4 Porous Adsorbent Materials, 3.3.5 Adsorption of Metals by Functionalized Silica-Based Adsorbent, 4. Conclusion.
  • General: On the whole, the text is a verbose style. You should revise the text.
  • Figure 1 and 4: It is difficult to understand these figures. You should add the detail explanation for these figures.
  • Application of Chelating Resin: You should add the information for functional structure and typical reaction using illustration to make the reader understand easily.
  • Affecting Adsorption: You should add the table for the roles of factors.
  • Table 8: You should delete no. and frame of “Carbon Nanomaterial based”.
  • General: You should check the subscript and superscript, e.g. “g/cm3”, ”OH-”, “H+”.
  • What is “CMB” without Abbreviation?

I recommended publication of this paper, subject to the above points being satisfactorily addressed.

Round 2

Reviewer 1 Report

The paper has been revised according to my suggestions and has reached the publication requirements. It is recommended to be published in this journal.